# Mind the Gap: Domain Gap Control for Single Shot Domain Adaptation for Generative Adversarial Networks

**Peihao Zhu**
KAUST

**Rameen Abdal**
KAUST

**John Femiani**
Miami University

**Peter Wonka**
KAUST

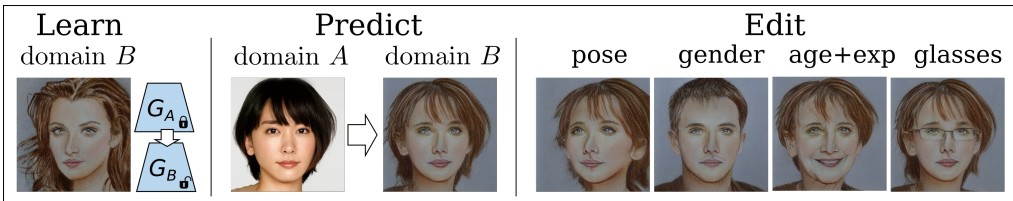

Figure 1: One-shot domain adaptation: (left) a single reference image from domain $B$ is used to refine a GAN $G_A$ to learn $G_B$; (center) every image in domain $A$ has an analog in domain $B$ that shares a latent code and many salient attributes; (right) because salient attributes are preserved in the new domain, many latent-edits are meaningful in the new domain.

## Abstract

We present a new method for one shot domain adaptation. The input to our method is a trained GAN that can produce images in domain $A$ and a single reference image $I_B$ from domain $B$. The proposed algorithm can translate any output of the trained GAN from domain $A$ to domain $B$. There are two main advantages of our method compared to the current state of the art: First, our solution achieves higher visual quality, e.g. by noticeably reducing overfitting. Second, our solution allows for more degrees of freedom to control the domain gap, i.e. what aspects of the image $I_B$ are used to define the domain $B$. Technically, we realize the new method by building on a pre-trained StyleGAN generator as GAN and a pre-trained CLIP model for representing the domain gap. We propose several new regularizers for controlling the domain gap to optimize the weights of the pre-trained StyleGAN generator so that it will output images in domain $B$ instead of domain $A$. The regularizers prevent the optimization from taking on too many attributes of the single reference image. Our results show significant visual improvements over the state of the art as well as multiple applications that highlight improved control[1].

## 1 Introduction

We propose a new method for domain adaptation based on a single target image. As shown in Fig. 1, given a trained GAN for domain $A$, and a single image $I_B$ from domain $B$, our approach learns to find a corresponding image in domain $B$ for any image in domain $A$. We can achieve this by fine-tuning the GAN for domain $A$ to obtain a second GAN that generates images in domain $B$. The two GANs share a latent space so that a single latent code will generate two corresponding images, one in domain $A$ and one in domain $B$. The main selling point of our method is that it achieves superior quality than the state of the art in single shot domain adaption. Our method is computationally lightweight and only takes a few minutes on a single GPU, so that it can be widely applied.

In order to do this, we leverage multiple existing components, including two excellent pre-trained networks: First, we use StyleGAN2 (Karras et al., 2020b) as a pre-trained GAN. A follow-up version has been published on arXiv (Karras et al., 2021), but the code only became available after we

---

[1]Code is available at `https://zpdesu.github.io/MindTheGap`.

finished all experiments. Second, we use a pre-trained network for image embedding, CLIP (Radford et al., 2021), to encode images as vectors. Third, we use the pioneering idea of StyleGAN-NADA (Gal et al., 2021), which builds upon StyleCLIP (Patashnik et al., 2021), to encode a domain gap (or domain shift) as vector in CLIP embedding space. Fourth, we leverage II2S (Zhu et al., 2020b) as GAN embedding method to transfer image $I_B$ into domain $A$ to obtain a better estimation of the domain gap.

Even though the visual quality of StyleGAN-NADA is already impressive when used as a single image domain adaption method, we identified multiple technical issues that can be improved to achieve another large jump in visual quality. First, and most importantly, StyleGAN-NADA was designed for zero-shot domain adaptation, and does not have a good solution to model the domain gap based on a single example image. Their reference implementation models the domain gap as a vector from the average image in domain $A$ to the given image $I_B$ in CLIP embedding space. However, this leads to overfitting in practice and the transfer results lose attributes of the input images, so that input images from domain $A$ get mapped to images that are all too similar to $I_B$ in domain $B$. We identify a better solution to this problem. In fact, the domain gap should be modeled as a vector from the image $I_B$ to its analog in domain $A$, so that the image in domain $A$ shares salient within-domain attributes with the reference image. We therefore need to solve an inverse $B$-to-$A$ domain-transfer problem, which we propose to tackle using the state-of-the-art GAN embedding method II2S (Zhu et al., 2020b). A key insight is that we can use a heavily regularized version of the II2S GAN inversion method to do the reverse problem of transferring any related image (from a domain $B$) into the domain $A$, helping to characterize the semantic domain gap better than previous work. Further extensions enable us to fine tune the modeling of the domain gap to explicitly model which attributes of the input image should be kept. Second, we propose multiple new regularizers to improve the quality. Third, we propose a technical improvement to the heuristic layer selection proposed in StyleGAN-NADA that is more straightforward and robust.

In summary, we make the following contributions:

1. We reduce the mode collapse/overfitting problem which often occurs in one-shot and few-shot domain adaptation. Our results look similar to the target domain images with fewer artifacts. These results are also faithful to the identities of the source domain images and able to capture fine details.

2. Our domain adaptation provides more freedom to control the "similarity" between images across domains that share a common latent-code, which makes a large number of downstream applications possible, e.g., pose adaptation, lighting adaptation, expression adaptation, texture adaptation, interpolation, and layer mixing, using state-of-the-art image editing frameworks.

## 2 RELATED WORK

**Domain adaptation.** Domain adaptation is the task of adapting a model to different domains. Different works in this area (Bousmalis et al., 2016; 2017; Na et al., 2020; Wang & Breckon, 2020; Kang et al., 2019) try to learn diverse domain independent representations using the source domain to make predictions, such as image classification, in the target domains. More importantly, generating diverse representations of images by combining natural language supervision has been of interest to the computer vision and NLP research communities (Frome et al., 2013). Recently, OpenAI's Contrastive Language-Image Pretraining (CLIP) (Radford et al., 2021) work established that transformer, and large datasets, could generate transferable visual models. In CLIP, both images and text are represented by high dimensional semantic-embedding vectors, which can then be used for zero-shot learning.

**GAN-based domain adaptation.** In the GAN domain, various models and training strategies have been proposed for few-shot domain adaptation tasks (Bousmalis et al., 2017; ZHANG et al., 2018; Li et al., 2020; Liu et al., 2019). Most relevant to our work, the domain adaptation methods (Patashnik et al., 2021; Gal et al., 2021; Jang et al., 2021; Song et al., 2021) that build upon StyleGAN (Karras et al., 2019; 2020b;a) demonstrate impressive visual quality and semantic interpretability in the target domain. These methods can be broadly classified into few-shot and single-shot domain adaptation methods.

A notable few-shot method, StyleGAN-ADA (Karras et al., 2020a) proposes an adaptive discriminator augmentation method to train StyleGAN on limited data. Another work, DiffAug (Zhao et al., 2020), applies differentiable transformations to the real and generated images for robust training. A discriminator related approach, FreezeD (Mo et al., 2020), freezes lower layers of the discriminator to achieve domain adaptation. Toonify (justinpinkney/toonify) interpolates between the model-weights of different generators to generate samples from a novel domain. A more recent work (Ojha et al., 2021), reduces overfitting on limited data by preserving the relative similarities and differences in the instances of samples in the source domain using cross domain correspondence.

**Latent space interpretation and semantic editing.** GAN interpretation and understanding of the latent space has been a topic of interest since the advent of GANs. Some notable works in this domain (Bau et al., 2018; 2019; Härkönen et al., 2020; Shen et al., 2020; Tewari et al., 2020a) have led to many GAN-based image editing applications. More recent studies into the activation space of StyleGAN have demonstrated that the GAN can be exploited for downstream tasks like unsupervised and few-shot part segmentation (Zhang et al., 2021; Tritrong et al., 2021; Abdal et al., 2021a; Collins et al., 2020; Bielski & Favaro, 2019), extracting 3D models of the objects (Pan et al., 2021; Chan et al., 2020) and other semantic image editing applications (Zhu et al., 2021; Tan et al., 2020; Wu et al., 2020; Patashnik et al., 2021).

Image embedding is one of the approaches used to study the interpretability of the GANs. To enable the semantic editing of a given image using GANs, one needs to embed/project the image into its latent space. Image2StyleGAN (Abdal et al., 2019) embeds images into the extended StyleGAN space called $W+$ space. Some followup works (Zhu et al., 2020a; Richardson et al., 2020; Tewari et al., 2020b) introduce regularizers and encoders to keep the latent code faithful to the original space of the StyleGAN. Improved-Image2StyleGAN (II2S) (Zhu et al., 2020b) uses $P_N$ space to regularize the embeddings for high-quality image reconstruction and image editing. We use this method to embed real images into the StyleGAN and show that our domain adaptation preserves the properties of the original StyleGAN in Sec 4.

Image editing is another tool to identify the concepts learned by a GAN. In the StyleGAN domain, recent works (Härkönen et al., 2020; Shen et al., 2020; Tewari et al., 2020a; Abdal et al., 2021b) extract meaningful linear and non-linear paths in the latent space. InterfaceGAN (Shen et al., 2020) finds linear directions to edit latent-codes in a supervised manner. On the other hand, GANSpace (Härkönen et al., 2020) extracts unsupervised linear directions for editing using PCA in the $W$ space. Another framework, StyleRig (Tewari et al., 2020a), maps the latent space of the GAN to a 3D model. StyleFlow (Abdal et al., 2021b) extracts non-linear paths in the latent space to enable sequential image editing. In this work, we will use StyleFlow to test the semantic editing of our domain adapted images.

In the area of text-based image editing, StyleCLIP (Patashnik et al., 2021) extends CLIP to perform GAN-based image editing. StyleCLIP uses the CLIP embedding vector to traverse the StyleGAN manifold, by adjusting the latent-codes of a GAN, in order to make a generated image's CLIP embedding similar to the target vector, while remaining close to the input in latent space. A downside to this approach is that these edits are unable to shift the domain of a GAN outside its original manifold. However, their use of CLIP embeddings inspired StyleGAN-NADA (Gal et al., 2021), which creates a new GAN using refinement learning to do zero-shot domain adaptation. Although unpublished, they also demonstrate one-shot domain adaptation in their accompanying code. The original and target domain are represented by CLIP text embeddings. The difference of the embeddings represents a direction used to shift the domains. Although in the accompanying source-code (rinongal/StyleGAN NADA), they use a bootstrap-estimate of the mean CLIP image embedding of the original domain, and use a reference image or its CLIP image embedding to represent the new domain.

## 3 METHOD

Our approach involves fine-tuning a GAN trained for some original domain $A$, e.g. FFHQ faces, to adapt it to a new related domain $B$. In our approach, the images in $A$ and the images in $B$ are related to each-other by a common latent code. Any image which can be generated or embedded in domain $A$ can be transferred to a corresponding and similar image in $B$. We use the CLIP

embeddings as a semantic-space in order to model the difference between domains $A$ and $B$, and we use StyleGAN (Karras et al., 2018; 2020b) as the image generator. A key to our approach is to preserve directions within and across domains as illustrated in Fig. 3. Before fine-tuning the GAN for domain $A$ (to obtain the GAN for domain $B$), we determine a domain-gap direction. This direction, called $v^{\text{ref}}$, is a vector in CLIP embedding space which points towards a reference image $I_B$ which is in domain $B$ from its corresponding image $I_A$ in which is in domain $A$. We use the CLIP image-embedding model $E_I$ to find

$$v^{\text{ref}} = E_I(I_B) - E_I(I_A). \tag{1}$$

Finding $I_A$ in domain A for a given image in domain B is a significant limitation in the current state of the art, StyleGAN-NADA (Gal et al., 2021), as they use the mean of domain $A$. The mean of domain A is a very crude approximation for $I_A$. Instead, we propose an inverse domain adaption step, by projecting the image $I_B$ into the domain $A$ to find a sample that is more similar and specific to the reference image than the mean of domain $A$. In principle, this problem is also a domain adaption problem similar to the problem we are trying to solve, just in the inverse direction. The major difference is that we have a pre-trained GAN available in domain A.

We use the II2S GAN-inversion method (Zhu et al., 2020b) in order to find a latent code for an image similar to $I_B$ that is plausibly within domain $A$. The I2S and II2S methods use an extended version of $W$ space from StyleGAN2. The $W$ code is used 18 times, once for each style block in StyleGAN2. When allowing each element to vary independently, the resulting latent space is called $W+$ space Abdal et al. (2019; 2020); Zhu et al. (2020b). I2S showed that the additional degrees of freedom allow GAN inversion for a wider set of images with very detailed reconstruction capabilities, and II2S showed that an additional regularization term to keep the latent codes close to their original distribution made latent-code manipulation more robust. II2S uses a hyperparameter, $\lambda$, which can be increased in order to generate latent codes using more regularization, and therefore in higher density regions of the $W+$ latent space. The effect of this parameter is shown in Fig. 2. The value suggested in the original work was $\lambda = 0.001$, however, low values of lambda allow II2S to find latent codes that are too far away from the latent-codes produced by the mapping network of the original GAN and thus produce images that are less plausible to have come from domain $A$, underestimating the gap between domains. In the context of domain shift we find it is useful to use $\lambda = 0.01$ as illustrated in Fig. 2. The result is a latent code $w^{\text{ref}}$ in $W+$ space which is shifted towards a high-density portion of the domain $A$. Then the image generated from that code, $I_A$, is an image in domain $A$ that corresponds to $I_B$.

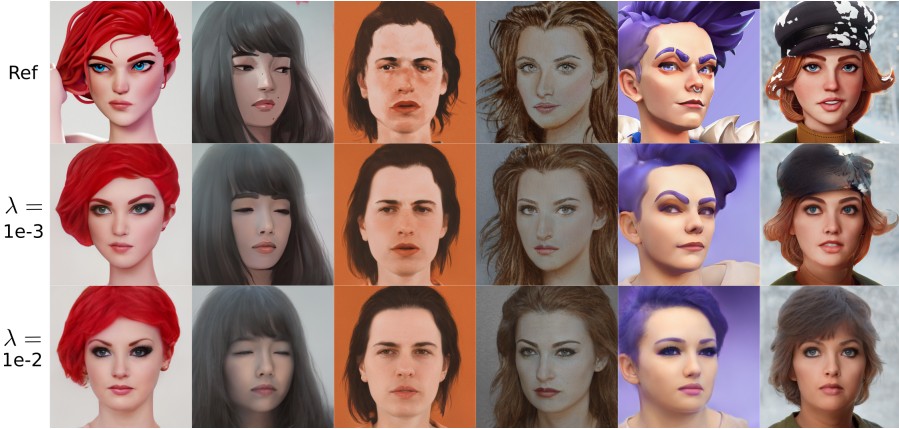

Figure 2: An illustration showing how II2S embeds $I_B$ in the original StyleGAN domain $A$, shown for two different values of $\lambda$. Reference images from other domains are shown in the top row. The value recommended by Zhu et al. (2020b) is shown in the second row, and the value used in this work is shown in the third row. Although there is some subjectivity involved, we believe that the large value $\lambda = 1\text{e}{-}2$ is needed for II2S to find images that plausibly could belong to the domain $A$, which in this case is FFHQ faces.

**Training**  As illustrated in Fig. 2, we use II2S to find an image $I_A$ which we consider to be similar to $I_B$ but still plausibly within a domain $A$. In principle, it is possible that II2S finds $I_A$ so that $I_B$ is similar enough to be considered the same, in which case the two domains overlap. However, we are concerned with the cases where the domains are different, and the vector $v^{\text{ref}}$ indicates the direction of a gap, or shift, between domain $A$ and domain $B$. We use refinement learning to train a new generator, $G_B$, so that images generated from $G_B$ are shifted parallel to $v^{\text{ref}}$ in CLIP space, relative to images from $G_A$. The desired shift is indicated by the red arrows in Fig. 3. During training, latent codes $w$ are generated using the mapping network of StyleGAN2. Both $G_A$ and $G_B$ are used to generate images from the same latent code, but the weights of $G_A$ are frozen and only $G_B$ is updated during training. The goal of refinement learning is that $G_B$ will preserve semantic information that is *within* domain $A$ but also that it will generate image shifted *across* a gap between domains. When refining the generator for domain $B$, we freeze the weights of the StyleGAN2 'ToRGB' layers, and the mapping network is also frozen. The overall process of training is illustrated in Fig. 4.

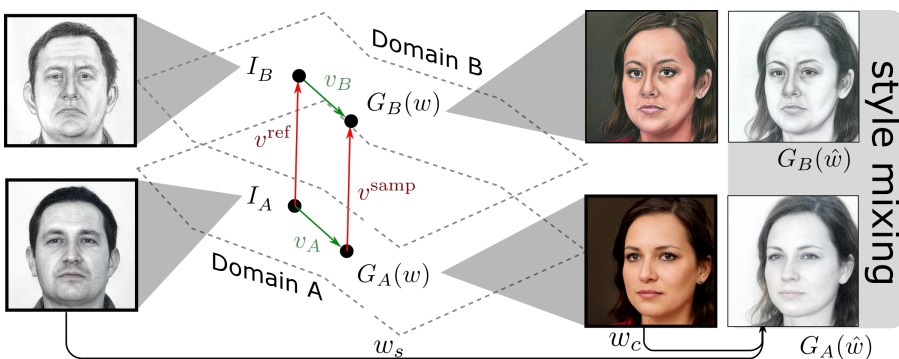

Figure 3: The vectors in the CLIP image embedding space, $E_I$, which control domain adaptation. Each domain is depicted here as a dashed outline; the vectors $v^{\text{ref}}$ and $v^{\text{samp}}$ cross between the two domains and are used to refine a generator for domain $B$. Corresponding images should be shifted in the same direction. The vectors $v_A$ and $v_B$ model important semantic differences within each domain that should also be preserved by domain transfer. $G_A(w)$ and $G_B(w)$ are corresponding images for an arbitrary latent-code $w$ encountered during training. Style mixing (shown on the right) shifts a part of the latent code towards the reference image effecting the result in both domains.

The goal of training is to shift CLIP embeddings from domain $A$ in a direction parallel to $v^{\text{ref}}$. We use the vector $v^{\text{samp}}$ to represent the current domain shift of the network $G_B$ during training, on a single sample. We have

$$v^{\text{samp}} = E_I(G_B(w)) - E_I(G_A(w)) \tag{2}$$

as a cross-domain vector for corresponding images generated from the same $w$ latent code using the two generators. We use the loss

$$L_{\text{clip\_across}} = 1 - \text{sim}(v^{\text{ref}}, v^{\text{samp}}), \tag{3}$$

where $\text{sim}(\mathbf{a}, \mathbf{b}) = \frac{\mathbf{a}^T \mathbf{b}}{\|\mathbf{a}\| \|\mathbf{b}\|}$ is the cosine similarity score. This loss term is minimized when the domain shift vectors are parallel.

It is important that the reference image $I_B$ matches the generated image, $G_B(w^{\text{ref}})$, both in a semantic sense, as measured by the similarity of the CLIP embeddings, and also in a visual sense. We accomplish this using two losses: $L_{\text{ref\_clip}}$ and $L_{\text{ref\_rec}}$. The first loss measures the change in the CLIP-embeddings of the original and reconstructed reference image,

$$L_{\text{ref\_clip}} = 1 - \text{sim}\left(E_I\left(I_B\right), E_I\left(G_B(w^{\text{ref}})\right)\right), \tag{4}$$

ensuring that the $G_B$ can reconstruct the embedding. Unlike $L_{\text{clip\_accross}}$, this loss term is not based on a *change* in embeddings between the two domains, instead it guides $G_B$ by aligning it to a global embedding in CLIP space, ensuring that $I_B$ remains fixed in the domain of $G_B$.

The second loss term is a reconstruction loss based on perceptual and pixel-level accuracy,

$$L_{\text{ref\_rec}} = L_{\text{PIPS}}\left(I_B, G_B(w^{\text{ref}})\right) + L_2\left(I_B, G_B(w^{\text{ref}})\right) \tag{5}$$

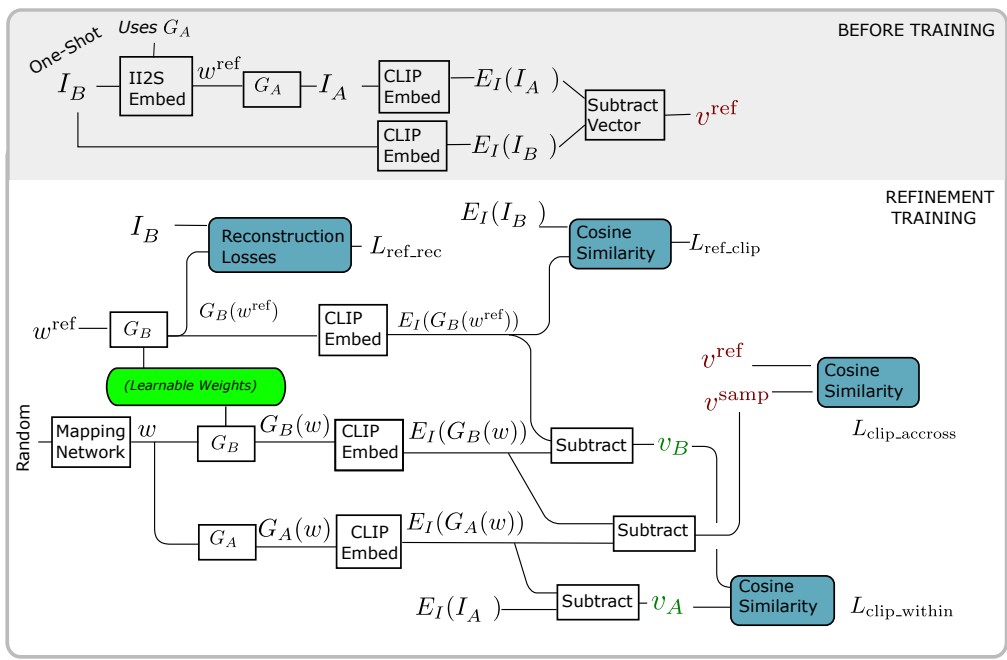

Figure 4: A process diagram for domain transfer. White rectangles indicate calculations, computed values are shown on the connecting lines. The four loss-calculations are indicated by blue rectangles, and the learnable weights of StyleGAN2 (all weights except the mapping network and the ToRGB layers) are indicated in green.

where $L_{\text{PIPS}}$ is the perceptual loss from Zhang et al. (2018), and $L_2$ is the squared euclidean difference between pixels. The purpose of this loss is to ensure that the visual, and not just the semantic, qualities of the image are preserved. This is necessary in addition to $L_{\text{ref\_clip}}$ because, while the CLIP embeddings do capture many semantic and visual qualities of the image, there are still many perceptually distinct images that could produce the same CLIP embedding. This is visible in Fig. 6, without the reconstruction loss $G_B$ fails to preserve some important visual qualities (such as symmetry) of the input.

There is a tendency for GANs to reduce the variation during training, especially in few-shot fine-tuning. We combat this by preserving the semantic information that is *not* related to the domain gap. A semantic change that is not related to the change in domains should not be affected by $G_B$. Therefore, the vector connecting the reference and sample images within the domain $A$ should be parallel to the corresponding vector in domain $B$. Let $v_A = E_I(G_A(w)) - E_I(I_A)$ be a vector connecting a sample image with latent-code $w$ to the reference image in the CLIP space. This vector represents semantic changes that are *within* domain $A$, and we want the matching semantic changes to occur within the domain $B$. Let $v_B = E_I(G_B(w)) - E_I(I_B)$ denote the corresponding vector in domain $B$. We introduce the loss

$$L_{\text{clip\_within}} = 1 - \text{sim}(v_A, v_B), \tag{6}$$

which is minimized when the two within-domain changes are parallel.

The final loss is then a weighted sum of losses

$$L = L_{\text{clip\_across}} + \lambda_{\text{clip\_within}} L_{\text{clip\_within}} + \lambda_{\text{ref\_clip}} L_{\text{ref\_clip}} + \lambda_{\text{ref\_rec}} L_{\text{ref\_rec}}, \tag{7}$$

with empirically determined weights of $\lambda_{\text{clip\_within}} = 0.5$, $\lambda_{\text{ref\_clip}} = 30$, and $\lambda_{\text{ref\_rec}} = 10$. Together, these four loss terms guide the refinement process for $G_B$. Among these losses, $L_{\text{clip\_across}}$ was proposed by StyleGAN-NADA (Gal et al., 2021). The other losses are novel contributions of this work.

**Style Mixing**  After the training step, the generator $G_B$ generates images that are semantically similar to the reference image $I_B$. However, we have observed that the visual style may not be

sufficiently similar. We attribute this to the idea that the target domain may be a *subset* of the images produced by the new generator $G_B$. This issue was addressed in StyleGAN-NADA (Gal et al., 2021) using a second latent-mining network in order to identify a distribution of latent codes within the domain of $G_B$ that better match the reference image. Our approach exploits the structure of latent codes in $W+$ space. Latent vectors in $W+$ space can be divided into 18 blocks of 512 elements, each impacting a different layer of StyleGAN2. Empirically, the latter blocks of the $W+$ code have been shown to have more effect on the style (e.g. texture and color) of the image whereas the earlier layers impact the coarse-structure or content (Zhu et al., 2021) of the image. We partition the latent code in the image into $w = (w_C, w_S)$ where $w_C$ consists of the first $m$ blocks of the $W+$ latent code that capture the content of the image, and $w_S$ consists of the remaining blocks and captures the style. In this work, we will use $m = 7$ unless otherwise specified. Then we transfer the style from a reference image using linear interpolation, to form $\hat{w} = (w_C, \hat{w}_S)$ where

$$\hat{w}_S = (1 - \alpha)w_S + \alpha(w_S^{\text{ref}}), \tag{8}$$

and $w_S^{\text{ref}}$ is last $(18 - m)$ blocks of $w^{\text{ref}}$. Consider the distribution of images generated from random $w$ drawn according to the distribution of latent codes from the mapping network of StyleGAN2. If $\alpha = 0$, then the distribution of images $G_B(\hat{w})$ includes the reference image, but encompasses a wide variety of other fine visual styles. If $\alpha = 1$, then the images $G_B(\hat{w})$ will still have a diverse content, but they will all very closely follow the visual style of $I_B$. An important application of this method is in conditional editing of real photographs. To achieve that, first we take a real input image $I_{\text{real}}$ and invert it in domain $A$ using II2S on the generator $G_A$ in order to find a $W+$ latent code $w_{\text{real}}$. Then $G_B(w_{\text{real}})$ generates a corresponding image in domain $B$. We can then compute $\hat{w}_{\text{real}}$ by interpolating the style codes (8) so that the final image $G_B(\hat{w}_{\text{real}})$ is similar to $I_{\text{real}}$ but has both content and the visual style shifted towards domain $B$.

## 4 RESULTS

In this section, we will show qualitative and quantitative results of our work. The only other published method that accomplishes similar one-shot GAN domain adaptation which we are aware of is Ojha et al. (2021). They focus on few-shot domain adaptation, but they also demonstrate a capability to solve the one-shot problem. The most closely related work to our approach is StyleGAN-NADA (Gal et al., 2021), which is unpublished at the time of submission, however we compare to it as the main competitor. The paper mainly discusses zero-shot domain adaptation, but the approach can also accomplish one-shot domain adaptation, as demonstrated in their accompanying source-code. Moreover, it demonstrates impressive improvements over the state of the art and even beats many SOTA few-shot methods considering the visual quality. As our method can still significantly improve upon the results shown in StyleGAN-NADA, this underlines the importance of our idea in reducing overfitting. We compare against additional approaches in the appendix.

**Training and Inference Time.** Given a reference image, the training time for our method is about 15 minutes for 600 iterations on a single Titan XP GPU using ADAM as an optimizer with the same settings as Gal et al. (2021). We use a batch size of 4. At inference time, there are different applications. In a basic operation, GAN generated images can be transferred with a single forward pass through a GAN generator network, which works in 0.34 seconds. Considering a more advanced operation, where existing photographs are embedded into a GAN latent space, the additional embedding time has to be considered. This embedding time is only 0.22 seconds using e4e (Tov et al., 2021) and about two minutes using II2S (Zhu et al., 2020b).

**Visual Evaluation.** In Fig. 5, we show a comparison of our results on faces against the two most relevant competing methods – StyleGAN-NADA (Gal et al., 2021) and few-shot-domain-adaptation (Ojha et al., 2021). The results show that our method remains faithful to the original identity of the embedded images in domain $A$, while the other two methods suffer from overfitting, i.e., collapsing to narrow distributions which do not preserve salient features (for example the identity of a person). We show additional visual results in the supplemental materials, including results on cars and dogs and results for fine-tuning the domain adaptation.

**User Study.** We also perform a user study by collecting 187 responses from Amazon Mechanical Turk in order to compare the visual quality and the domain transfer capabilities of our framework

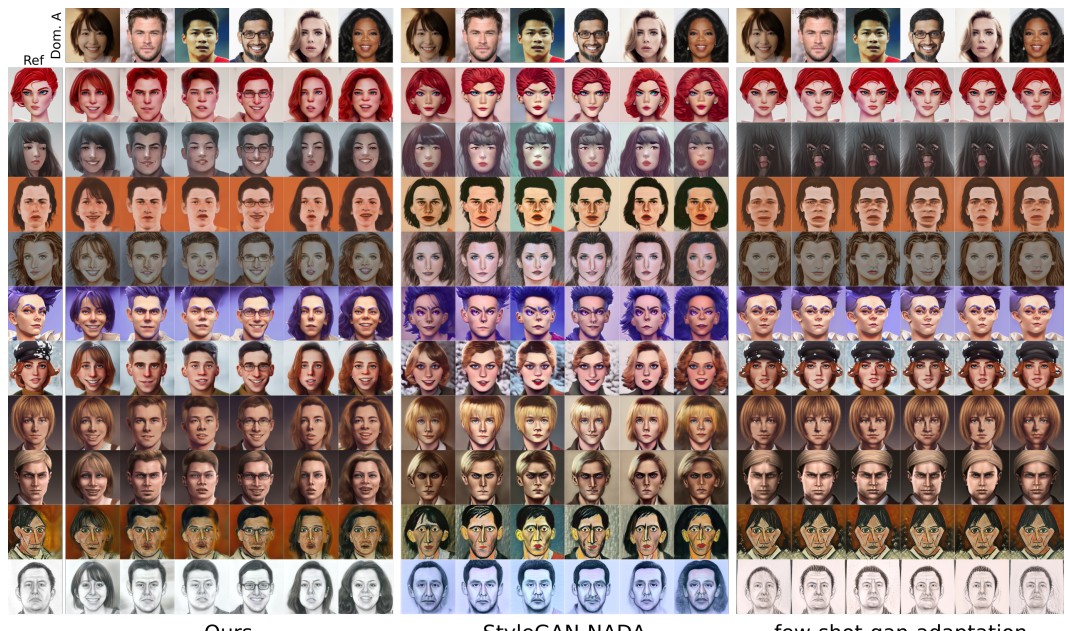

Figure 5: Comparison of our framework with state-of-the-art frameworks for StyleGAN domain adaptation. We compare with StyleGAN-NADA (Gal et al., 2021) and the few-shot method of Ojha et al. (2021). Each row corresponds to a different reference image $I_B$, and each column is a different real image $I_{\text{real}}$ from domain $A$. Notice that our method is able to match the styles of the reference images, while StyleGAN-NADA fails to maintain the content of the images in domain $A$ (for example the identity of a person is lost). On the other hand, the few-shot method suffers from severe mode collapse.

compared to the competing methods. When asked which method generates higher quality images from domain $B$, 73% of users preferred our approach to StyleGAN-NADA, and 77% selected ours over Few-shot (Ojha et al., 2021). When asked which method is better at maintaining the similarity to a corresponding source image in domain $A$, we found that 80% of the responses chose our approach over StyleGAN-NADA, and 91% preferred our approach to Few-shot. Our method outperforms the competing works in terms of the quality of the generated image, and the similarity of the generated image to the source image from domain $A$. According to the user study, the other methods produced images that are more similar to $I_B$, but that is also an indication of overfitting and mode collapse.

**Ablation study.** We perform an ablation study to evaluate each component of our framework. In Fig. 6, we show the effect of II2S embedding, different losses and style mixing/interpolation on the output.

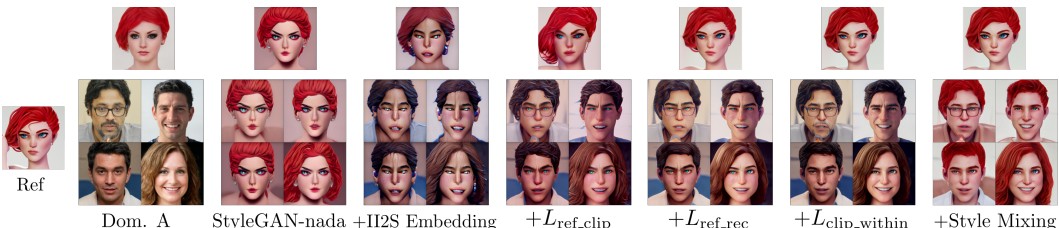

Figure 6: Ablation study of the losses and style mixing used in our framework. From left to right: the reference image $I_A$ and several images from domain $A$, the baseline approach (StyleGAN-NADA), adding II2S instead of using the mean of domain $A$, adding $L_{\text{ref\_clip}}$, $L_{\text{clip\_within}}$, and then using style mixing. The top row shows reconstructions of the image $I_A$ using $G_B$.

**Image editing capabilities.** Another important aspect of our method is that we are able to preserve the semantic properties of the original StyleGAN (domain $A$) in domain $B$. We can make edits to the images in domain $B$ via the learned generator $G_B$ without retraining the image editing frameworks on the new domain. Fig. 7 shows image editing capabilities in the new domain $B$. We use StyleFlow edits such as lighting, pose, gender etc. to show the fine-grained edits possible in the new domain.

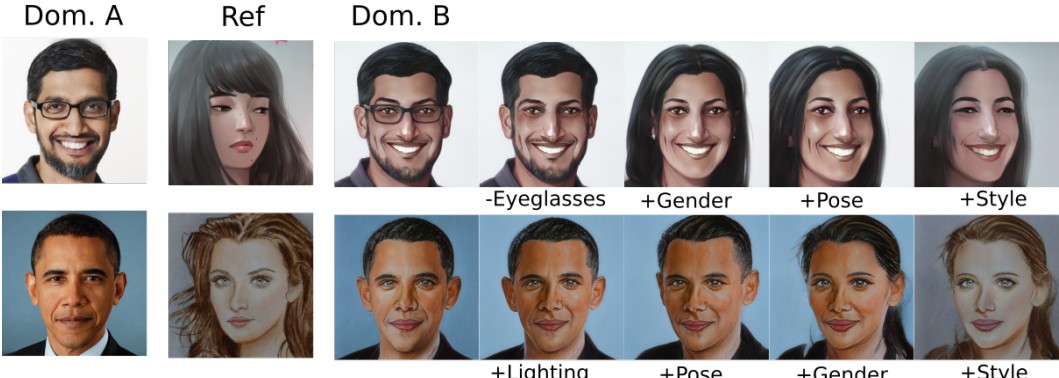

Figure 7: Image editing capabilities of the new domain $B$ using StyleFlow (Abdal et al., 2021b). This figure shows the editing results of the embedded real image $I_{real}$ transferred to domain $B$. Notice that our method preserves the semantic properties of the original StyleGAN.

**Limitations** Our method has several limitations (See Fig. 8). Some of these limitations are inherent due to the challenging nature of the problem of single-shot domain adaptation. Other limitations can be addressed in future work. First, when we find the initial image in domain $A$ that corresponds to the input in domain $B$, we do not attempt to control for the semantic similarity. Future work should encourage the images to have similar semantics. Second, we can only transfer between related domains. For example transferring FFHQ faces into the domain of cars is not explored in this paper. Third, also relevant to the original distribution of the StyleGAN, embeddings into the Style-GAN work best when the objects are transformed to the canonical positions (for example face poses that are the same as FFHQ). Extreme poses of the objects in the reference images sometimes fail.

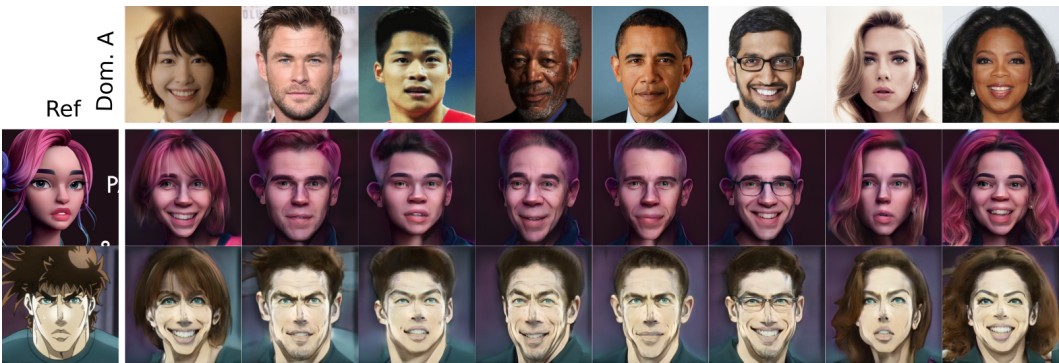

Figure 8: Some failure cases of our method. In these examples, we observe that the identity of the face is compromised a bit more than in typical examples of our method.

## 5 CONCLUSIONS

We propose a novel method for single shot domain adaption. The main achievement of this work is to obtain results of unprecedented quality while reducing overfitting observed in previous work. The technical key components of our work are a method to model the domain gap as vector in CLIP embedding space, a way to preserve within-domain variation, and several extensions for fine-grained attribute-based control. We also introduce several new regularizers and a style mixing approach.

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

# A    APPENDIX: ADDITIONAL RESULTS

## A.1    VISUAL EVALUATION OF STYLE TRANSFER.

We provide additional visual evaluation of the results. In Fig. 9 and 10 we show results of domain adaptation applied to faces. The input photographs are in the top row and the reference images are in the first column. We can see that the results take on the style of the reference image, even though the reference image is far outside the original GAN's latent space. Also, we notice that overfitting is successfully limited, as each result maintains several important aspects of the input image. In Fig. 13 and  14 we show results for cars, cats, and dogs on the same task. This shows that our method is consistent across different StyleGAN objects/datasets.

## A.2    QUANTITATIVE COMPARISON OF SKETCH IMAGES.

We calculate the FID (Heusel et al., 2017) between 1,000 generated images and the entire sketch dataset. Additionally, we report the precision and recall metric (Kynkäänniemi et al., 2019) to measure the quality and diversity respectively. As shown in Tab. 1, our method outperforms the contemporary methods Few-Shot (Ojha et al., 2021), and StyleGAN-NADA (Gal et al., 2021) on both metrics.

Another contemporary method, TargetCLIP (Chefer et al., 2021), is capable of one-shot 'essence transfer' using a latent-edit, however as the weights of the generator are not modified their approach is restricted to the manifold of the original generator. Because it cannot shift to a completely new domain, TargetCLIP failed to produce any sketch images and has a precision=0. Because the images it did generate are in the original space of StyleGAN it has high recall (0.29), but this number is not meaningful.

Unsurprisingly, all one-shot domain transfer methods have low recall (low diversity) but it is significant that ours is the only approach with positive recall to within 2 significant digits.

Table 1: Quantitative comparison on one-shot adaptation between few-shot-domain-adaptation, StyleGAN-NADA, and our method. Evaluation metrics include FID, precision, and recall (higher means higher diversity).

| One Shot Method | FID↓ | precision↑ | recall↑ |
|---|---|---|---|
| Few-shot (Ojha et al., 2021) | 158.86 | 0.00 | 0.00 |
| SG-NADA (Gal et al., 2021) | 124.55 | 0.12 | 0.00 |
| Ours | **78.35** | **0.33** | **0.02** |

## A.3    MULTI-SHOT DOMAIN ADAPTATION.

Although it was designed for one-shot domain adaptation, our method can be extended to few-shot domain adaptation by using multiple input/reference image pairs $(I_A, I_B)$. In Fig. 11, We show the visual improvement obtained using 3-shot reference images.

## A.4    CONTROLLING THE STYLE GAP

Our method provides a way to control the domain gap between the domain $A$ and domain $B$ by explicitly controlling the style of the images sampled from or embedded in domain $A$. Fig. 12 shows that we can control the degree to which style from the reference image is preserved by increasing the style-mixing parameter $\alpha$, which is not possible with any of the competing methods. This gives users more control over content generation and editing.

## A.5    ADDITIONAL COMPARISON

In addition to our comparison with StyleGAN-NADA (Gal et al., 2021) and few-shot domain adaptation (Ojha et al., 2021), we compare against three additional methods in Fig. 15. These include one concurrently developed method called TargetCLIP (Chefer et al., 2021) as well as two other methods that work on lower resolution images for one-shot domain transfer. These are the method

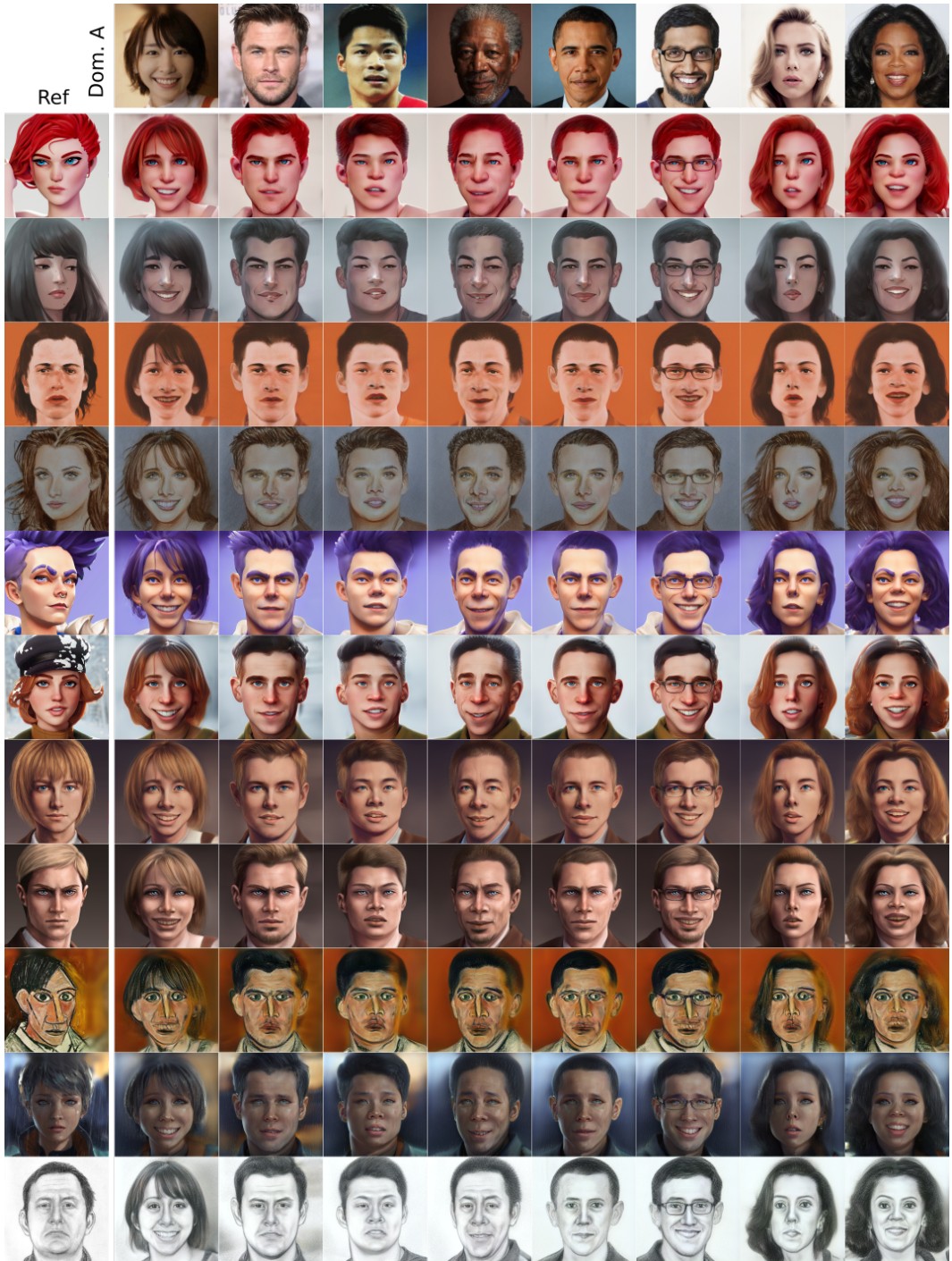

Figure 9: Style transfer results obtained by our method after style interpolation in domain $B$. The top row represents the real images embedded in the latent space of $G_A$ (domain $A$) whose latent codes are then used by $G_B$ (domain $B$). The first column represents the reference images $I_B$ which are input to our domain adaptation framework.

of Gatys *et. al* Gatys et al. (2016) and the the AdaIN approach (Huang & Belongie, 2017). Our visual results compare favorably against the new methods in Fig. 15 with respect to preserving the identity of the original image while also generating images that belong to the new domain.

## A.6 INFERENCE AND EDITING TIME

Our proposed approach uses II2S for training and inference and StyleFlow (Abdal et al., 2021b) for editing in the new domain. GAN inversion using II2S on HD ($1024 \times 1024$) images takes 150 seconds on average, and each latent-code edit operation takes 0.47 seconds. Generating the images afterwards takes an addition 0.34 seconds. Note that the run-time is dominated by GAN -inversion using II2S, however as we show in Fig. 16 once training is completed, we can use other GAN inversion methods to accomplish the edits. With e4e (Tov et al., 2021) inversion is only 0.22 seconds and the entire process of inversion, editing, and generating the edited image can be accomplished in approximately one second.

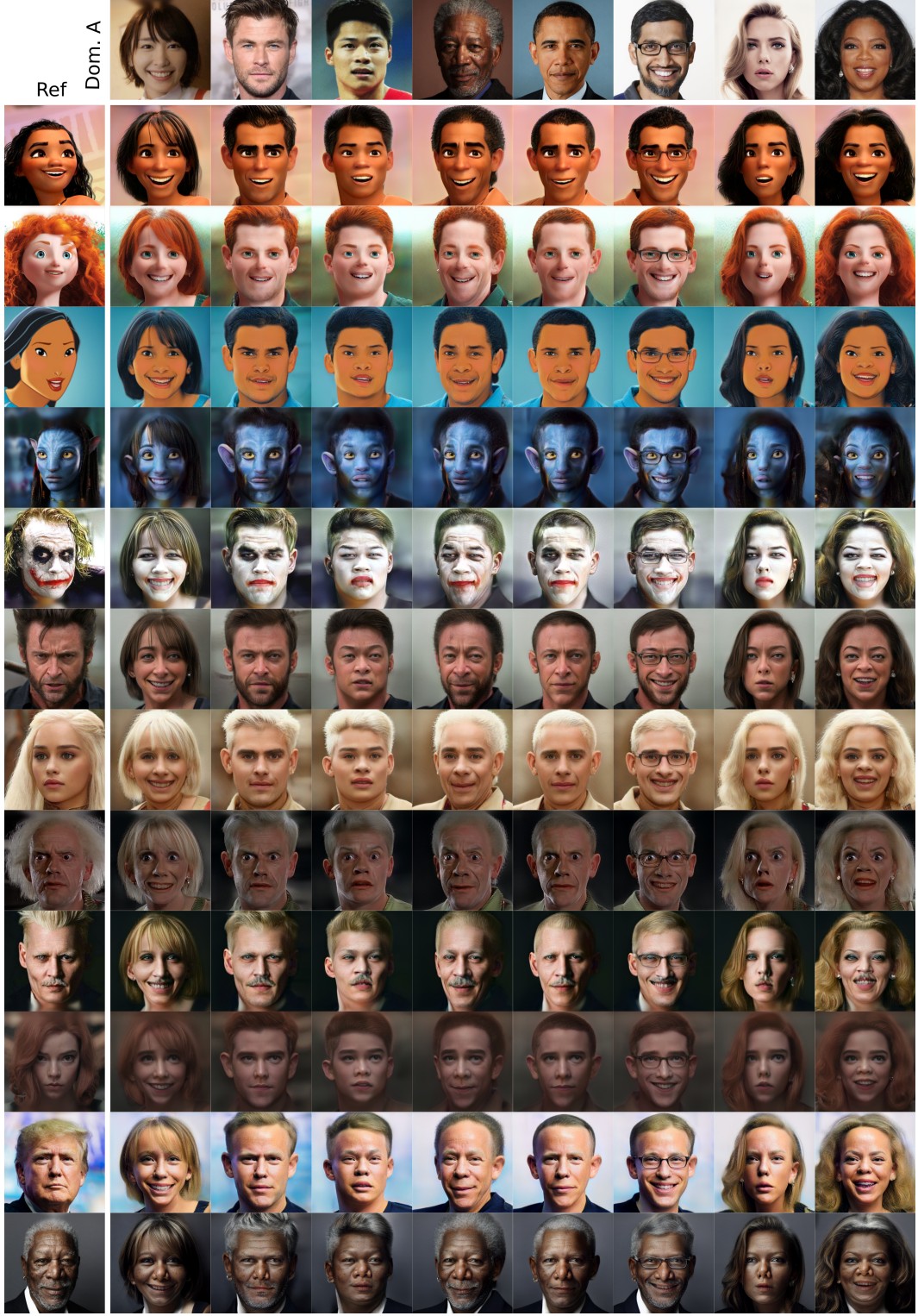

Figure 10: The structure of rows and columns is the same as in Fig. 9. Note: our method also works well when the reference images are real face images.

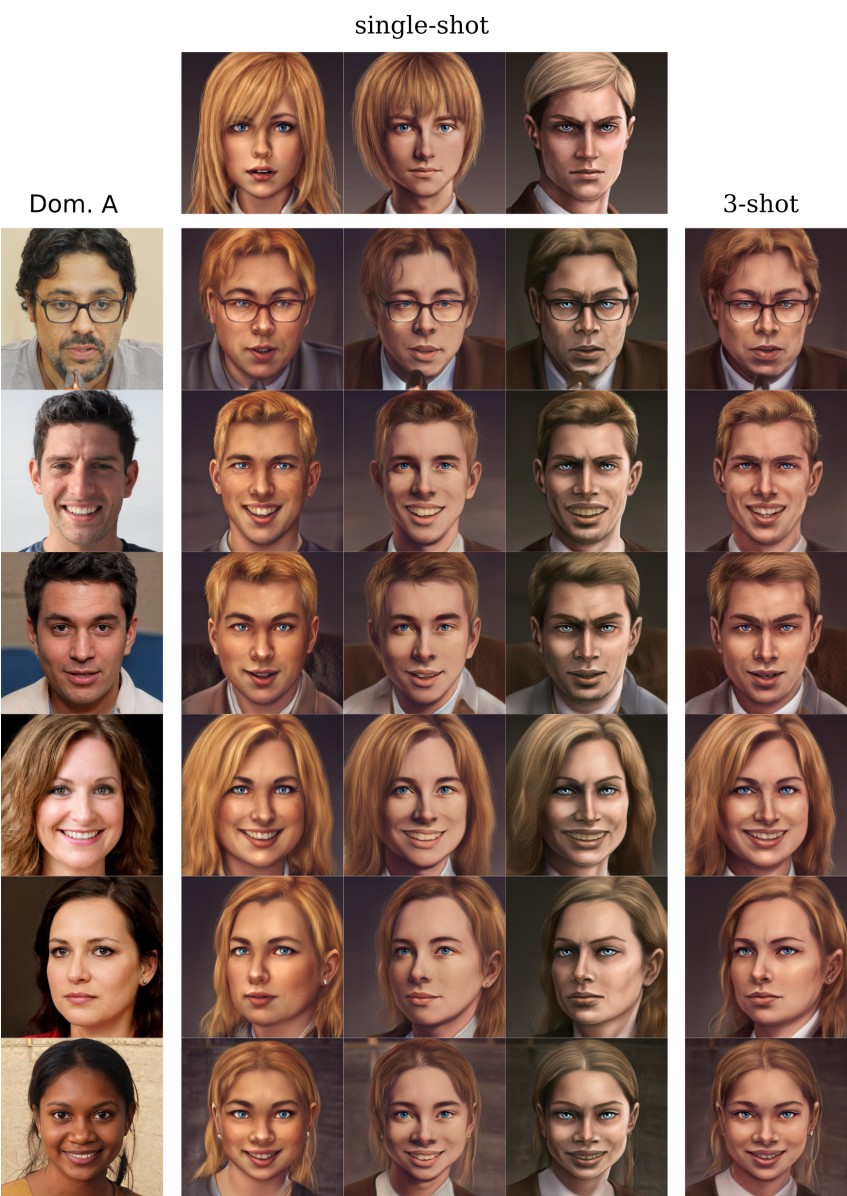

Figure 11: Our method extends to deal with multiple reference images. The figure compares the results using 3 reference images and using single reference image. It can be observed that our method can better catch the general style and achieve more stable results when given multiple reference images.

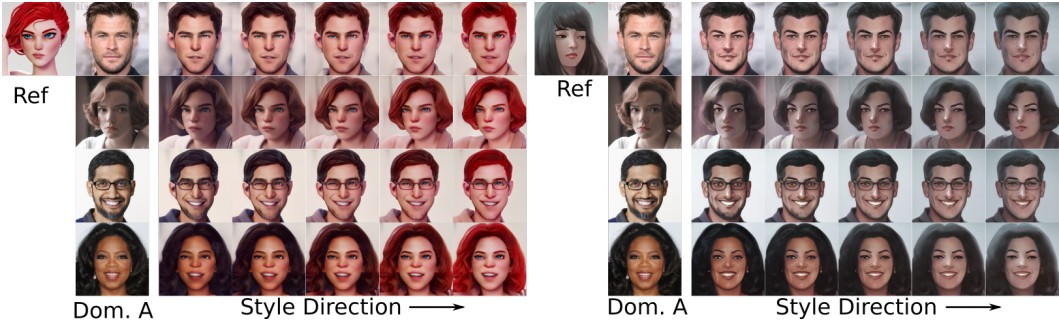

Figure 12: Style interpolation results achieved by our framework. Unlike the competing methods, our method has an explicit control over the styles in the domain $B$. Each sub figure shows a reference image and images embedded in domain $A$. Notice that we can control the amount of variation in style depending on a parameter $\alpha$ that can be specified by a user.

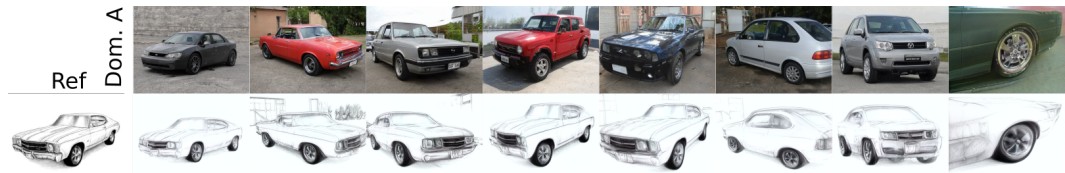

Figure 13: Our domain transfer results on cars. The structure of rows and columns is the same as in Fig. 9.

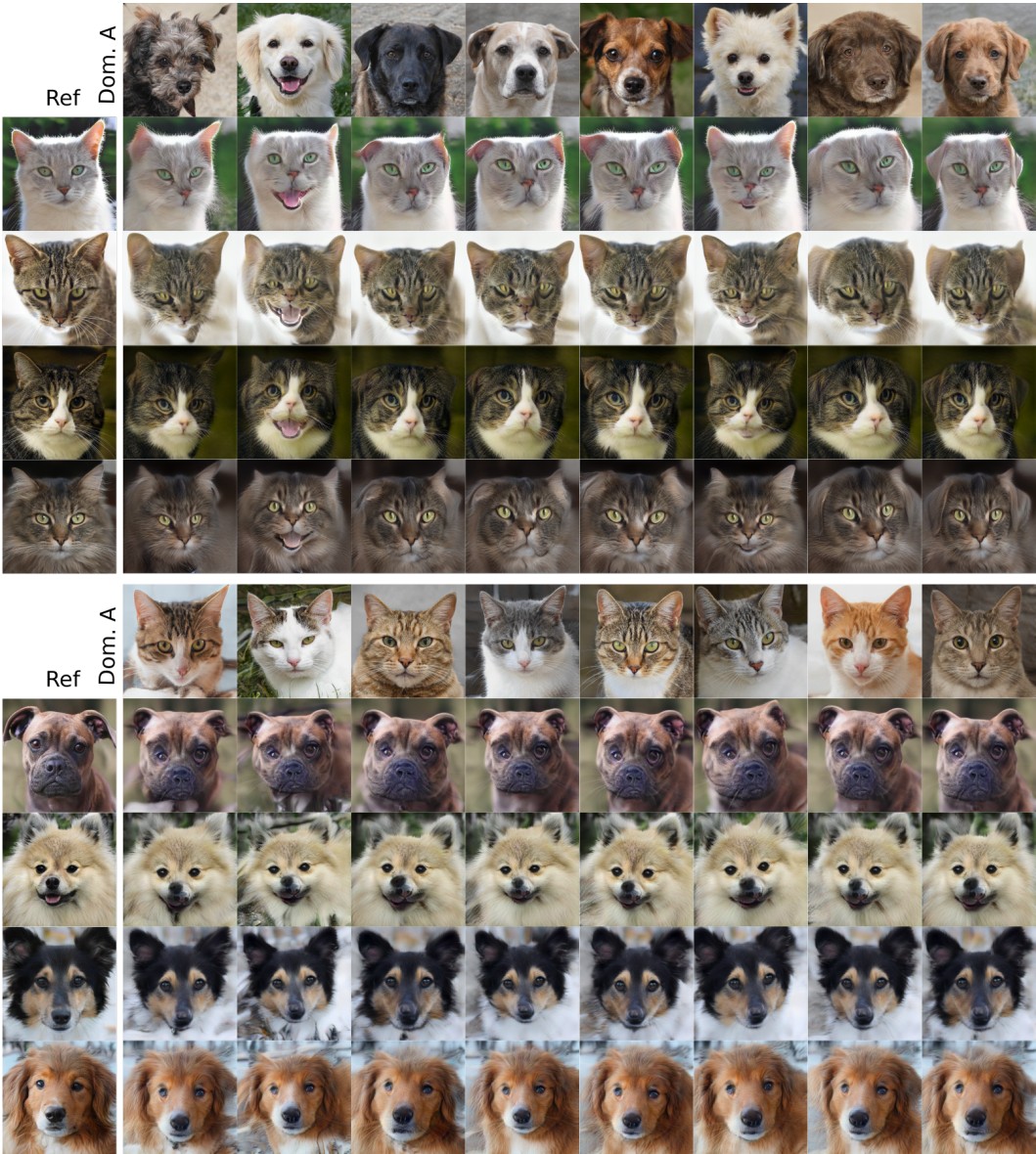

Figure 14: Our domain transfer results on cats and dogs. The structure of rows and columns is the same as in Fig. 9.

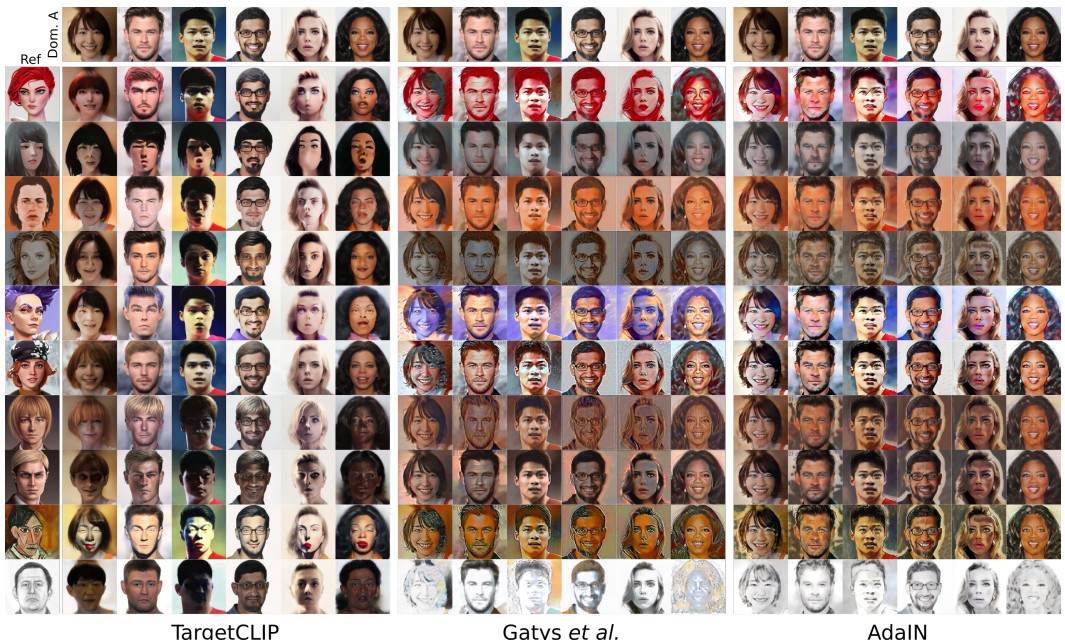

Figure 15: Additional comparisons with other baseline methods including the concurrent method TargetCLIP (Chefer et al., 2021) as well as two lower-resolution methods from Gatys et al. (2016) and AdaIN (Huang & Belongie, 2017). One-shot reference images from domain $B$ are shown in the left column. Each image is the result of transferring the image in the top row into the new domain. Compare these images to our method in Fig. 7, our proposed approach has fewer artifacts while preserving the identity of the image in domain $A$.

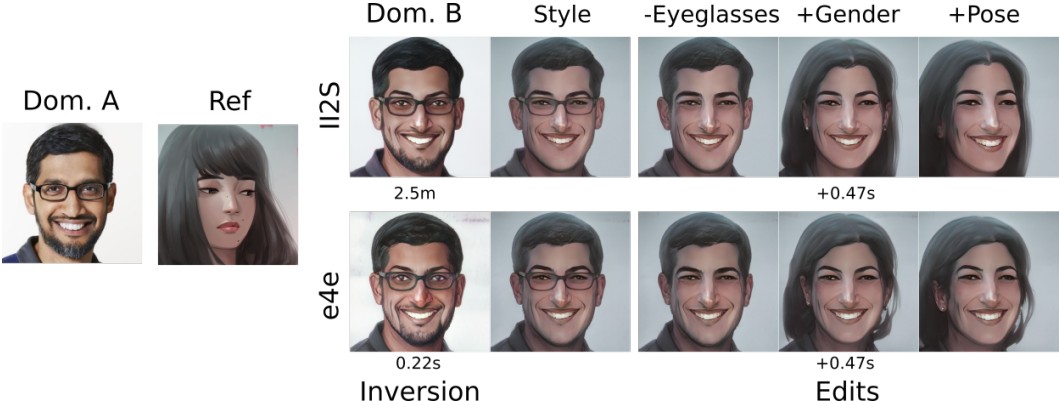

Figure 16: Comparison domain-transfer and editing using II2S vs e4e. The new GAN is always trained using II2S, but once training is complete, e4e can be used to transfer images into the new domain. II2S takes 2.5 minutes to embed the image, while e4e needs about 0.22 seconds. StyleFlow editing takes 0.47 seconds, and StyleGAN image generation takes about 0.34 seconds.

