# OpenReview forum: "Mind the Gap: Domain Gap Control for Single Shot Domain Adaptation for Generative Adversarial Networks"
_ICLR.cc/2022/Conference — ICLR 2022 Poster_

### Official Review · Reviewer_d3yx · 2021-11-01

**Correctness:** 2
**Technical Novelty And Significance:** 2
**Empirical Novelty And Significance:** 2
**Recommendation:** 6
**Confidence:** 4

**Main Review:**

Strengths:
1. This proposed method by combining StyleGAN and CLIP is elegant.
2. The experiments demonstrate the effectiveness of their method, and the visual results in the paper are promising.

Weakness:

1. About Figure 6, why use `+` before every semantics?  It is obvious that the glasses were removed from the man's face instead of adding. And what is the difference between the `Ref` and `Dom.B` in the bottom group?


==========Update==========

At first, the results in this paper left an impressive result in my mind, and then I gave a high score. However, after reading some papers that also have been submitted to ICLR2022, which do the same task, compared with them, I found some weaknesses in this paper and therefore changed my score.

1. The whole paper lacks quantitative results, and all the presented results are some cherry-picked images.  And for lacking comparisons with SOTA proposed by other reviewers, although you have added some, there is still a lack of quantitative comparison.
2. The main experiments are only style transfer on human faces and lack the attributes transfer. This task can not be called `generative domain adaption.`  Although some extra experiments on cars and cats are shown in the appendix, those results are also misleading. For instance, in figure 11, the caption says this is a style transfer task, but the dogs' structure changed vastly.
3. The diversity of the transferred model is not validated as well, which is an important index to evaluate the generative models.
4. When compared with other methods, using the CLIP model is kind of unfair.


**Summary Of The Paper:**

This paper mainly deals with domain transfer tasks in pre-trained GANs, such as transferring the generated images from domain A to domain B by a single reference image from domain B. And experiments show that the proposed method surpasses the SoAT.

**Summary Of The Review:**

The results in the paper are a little amazing and can be used in style transfer on human faces.

---

> ### Author Response · Authors · 2021-11-22
> **Response to Reviewer d3yx**
>
> **Q1: About Figure 6, why use + before every semantics? It is obvious that the glasses were removed from the man's face instead of adding. And what is the difference between the Ref and Dom.B in the bottom group?**
>
> R: The original idea is that we added the ability to change the feature (not that we added an object).  Some aspects of an image (such as gender) do not have a clear semantic relationship with the '+' sign, however we recognize that edits indicating the presence or lack of an object (like eyeglasses) vs the addition of an editing capability may cause confusion when a + sign is used. We have modified the figure in the paper by using '-' signs for certain features in the paper.
>
> The bottom portion of the figure will be removed from the main paper. We understand it was confusing because the same images were not aligned in the top and bottom portion of the figure, but also it was attempting to show something that is hard to convey and not critical to the flow of the paper.  In the original (now removed) portion of the figure, the top portion used a real image from domain A and found a corresponding generated image in domain B.  In the bottom portion, the real image is from domain B, it is the same as the Ref image, and a generated image in domain A is not shown nor is it needed to do the edit.  The ‘Dom B’ column showed a reconstruction of the Ref image using the generator for domain B.

---

> ### Author Response · Authors · 2021-11-28
> **?????**
>
>
> **We would like to note that the reviewer changed his score from 8 to 3. The strange thing is that:**
> * The review change happened on the 28th / 29th of November (depending on where on the world you are), just shortly before the end of the final discussion period.
> * While it is legitimate to change one's mind in principle, our rebuttal was posted on the 22nd. The timing of the change gives very little time for a discussion, because we would love to hear back from the reviewer about details.
> * Overall, the content of the new review is very unusual. The main content of the new review is that other submitted ICLR 2022 are allegedly better (without giving any references), have better results, and have better evaluation (without giving any details on how the results or evaluation is better). This is surprising, because our results are very strong. We also looked at other ICLR 2022 submissions and came to the opposite conclusion. We believe our method would do very well in comparison to concurrent work. We already compare to multiple other concurrent papers and are not aware of a better one submitted to ICLR 2022. We are happy to attempt any comparison to other ICLR 2022 submissions. It’s possible that we missed another relevant submissions.
>
> More details below:
>
> **Review:** At first, the results in this paper left an impressive result in my mind, and then I gave a high score. However, after reading some papers that also have been submitted to ICLR2022, which do the same task, compared with them, I found some weaknesses in this paper and therefore changed my score.
>
> **A:**  This does not seem to be ethical or fair and we urge the reviewer to reconsider their update. Our paper did not get worse and we addressed the reviewers issues. Without any references we don't know what concurrent ICLR submissions the reviewer is talking about and furthermore they do not have a bearing on the correctness of our work.  There is only one day of discussion left, and the reviewer has not indicated which papers we failed to compare with or which metrics are missing. Even compared to concurrent ICLR 2022 submissions, we feel our paper does very well.
>
> **Review:** The whole paper lacks quantitative results, and all the presented results are some cherry-picked images. And for lacking comparisons with SOTA proposed by other reviewers, although you have added some, there is still a lack of quantitative comparison.
>
> **A:** First, regarding “cherry picked” results.  The images were not cherry picked. In the figures we specifically selected a variety of race and gender, hairstyle, and glasses. We use more example target domains than prior art.  We show failure cases in Fig 8. It does not seem fair to call these cherry-picked results.
> Second, regarding quantitative results.  It is possible to quantitatively evaluate zero-shot classification but no such metric is proposed for GANs. We DO conduct a user study. There is NO numeric metric we are aware of to evaluate zero or one-shot domain adaptation for GANs.  We urge the reviewer to look at other single-shot GAN methods like AdaIN, StyleGAN-Nada, etc. to see how qualitative results and user studies are currently the best way we have to evaluate zero/one shot GAN. This is also the case with Style transfer problems. We appreciate the value of metrics however any metric one tries to invent will certainly fail to measure anything meaningful when applied one-shot generated images. We put careful thought into what quantitative measure would be useful - although some almost make sense in low-shot scenarios where the system has some indication of the variance of the target domain,  there is not much to do in a one-shot scenario. How much variance _should_ the domain-B generator have? Which attributes SHOULD be preserved? For example, when transferring dogs to cats, our method generates cats with floppy ears. This is possible but rare among cats, but that is not quantifiable. If an alien saw this it would be convinced by the results.
>
> **Review:** The main experiments are only style transfer on human faces and lack the attributes transfer. This task can not be called generative domain adaption. Although some extra experiments on cars and cats are shown in the appendix, those results are also misleading. For instance, in figure 11, the caption says this is a style transfer task, but the dogs' structure changed vastly.
>
> **A:** We used the wrong word in the caption -- that should say “domain” transfer, including style mixing. It is changing the structure as well as the style, as it should. We can also show the results without the style part described at the end of section 3.

---

> > ### Author Response · Authors · 2021-11-28
> > **????? Part-2**
> >
> > **Review:** The diversity of the transferred model is not validated as well, which is an important index to evaluate the generative models.
> >
> > **A:**  There is no way for a one-shot system to know the true distribution of the target domain. Based on our parameter ‘alpha’, we can generate models with more or less variance.  Fig.10 shows how increasing alpha reduces the variance and makes images look more like the reference image. Although we eliminated if for clarity & brevity, one can also truncate the other weights in the W+ code to reduce variance in structure. We did not ever do this in the experiments we currently present.  Not all diversity is the same -- variation must correspond to meaningful changes in the images. A number cannot capture this in 1-shot scenarios. A single image belongs to multiple potential domains and without knowing which it is our system cannot match any particular one exactly, nor could any system. This is why figures are needed. Our images have more diversity than competing methods based on StyleGAN, however this must be judged by inspecting our figures or using the code we will share.
> >
> > **Review:** When compared with other methods, using the CLIP model is kind of unfair.
> >
> > **A:**   So which is it -- we should compare to more prior art, but can’t use work that does not use CLIP?  CLIP came out this year,  we compare it with two methods that use CLIP, both are actually concurrent. In general it is not fair to expect authors to compare with concurrent work.

---

> > ### Comment · Reviewer_d3yx · 2021-11-29
> > **Response**
> >
> > 1. What does `?????` means? The deadline for CVPR is 25th. After that, I have a short rest and then start reading some papers on ICLR2022. And I realized that I might give a too high score on this paper. I do not find any unusual. And I think we still have time to discuss.
> >
> > 2. You **can not** tell others that your method performs very well by simply showing several images without giving any quantitative results since GANs can generate millions of results while every model (method) has a failure case. For a generative model, report FID and Precision and Recall curve is an essential requirement.

---

> > > ### Author Response · Authors · 2021-11-29
> > > **FID, Precision, and Recall**
> > >
> > > **Thanks for being willing to discuss.** FID, Precision, and Recall are used to compare two distributions of images. In our case, the domain B is given by a single image. For example, FID should have 50000 images, both synthetic and real. While we can generate 50000 images, we only have 1 image to compare to.
> > > Which other one-shot papers out there use FID, Precision, and Recall for the evaluation? As far as we know, the most important evaluation is a user study, and visual comparison. That is what we did.
> > > It would be really helpful for us if the reviewer can actually point out what other papers they are talking about.
> > > FID, Precision, and Recall was not computed in the other similar one-shot papers:
> > > StyleGAN-NADA: CLIP-Guided Domain Adaptation of Image Generators
> > > Image-Based CLIP-Guided Essence Transfer
> > >
> > > Still, the results of our method are so strong and a significant improvement over previous work. Even if a meaningful metric could be computed it would most likely show that our method is very good.
> > >
> > > It would be possible to make such a comparison, if another domain with 50,000 images could be found. For example, Ojha uses the domain of babies, sunglasses, and sketch. The domain of babies and sunglasses are too similar to the original domain. This would make the method more similar to attribute transfer or editing. The cases of babies and sunglasses are too easy and not indicative of challenging domain transfer problems.
> > > Sketches we could have done, but there are many problems as well. There are only 300 images available which makes the results very questionable. However, since Ojha has almost a complete mode collapse on sketches, our results are virtually guaranteed to be better in any metric someone can come up with.

---

> > > > ### Comment · Reviewer_d3yx · 2021-11-29
> > > > **Thanks for the authors' response**
> > > >
> > > > First of all, I would like to re-explain the reason for the score change. I change my mind, *not* because this work is not compared with concurrent work or even other ICLR submissions, but because **there lack some important comparisons as well as the quantitative evaluations**. The previous comments may cause some misunderstanding, for which I apologize. Here, I would like to correct the statement: **I just realize the missing of some comparisons and evaluations in this work *after reading some recent papers*, but *NOT* make the decision because of them**. Thanks for the authors reminding me of this incorrectness.
> > > >
> > > > Now, I want to discuss two major weaknesses I have recently found out. The authors are right that the discussion period is about to close and it may not be that fair to raise a problem at this time. But I still think **it is the responsibility of reviewers to discuss a paper thoroughly and seriously, whatever time it is**. I can agree that ACs do not take such a discussion into account when making the decision, but I **insist to express my thoughts and hope they can help improve this work in the future**.
> > > >
> > > > On one hand, the topic studied in this paper is **"domain adaptation for GANs"** (as in the title). But all presented results are more like "style transfer", which is more suitable to the task image-to-image translation. The **main** concern is that, **for a generative model, diversity is as (if not more) important as the quality**, because a generative model is expected to reproduce an observed distribution instead of only performing some editing tasks. From this perspective, I do not think "domain adaptation for GANs" is a proper description of this work. Ojha et al. and many already published papers **have already given adequate experimental settings in this task**, but this work does not conduct any of them.
> > > >
> > > > I am **not** convinced by the explanation "The domain of babies and sunglasses are too similar to the original domain. This would make the method more similar to attribute transfer or editing." On the contrary, from my viewpoint, the results shown in this submission are more like **style transfer or editing** and could **not** be called domain adaptation for GANs. **Furthermore, even "the domain of babies and sunglasses are too similar to the original domain", it will still be good to see how the proposed approach performs under such a setting.** Can it properly extract the variation of "wearing eyeglasses" or not? Only from the results in the submission, I cannot get a convincing conclusion. That is why I say **there lack some important comparisons**.
> > > >
> > > > On the other hand, without a model (or code) to test by myself, I am not convinced that qualitative results are not cherry-picked. For example, can the authors promise they have presented all results when doing experiments? Or synthesize 100 images and select the best 10? User study is an acceptable metric but still not that convincing, because it highly depends on the users. As I have mentioned above, **quality and diversity** are two important criteria to evaluate a generative model, **both of which have well-defined evaluation protocol in the literature**, as FID and precision-recall. I do **not** buy the statement "There are only 300 images available which makes the results very questionable." **Why can previous works, like Ojha et al., perform quantitative evaluation, but cannot this work?** In addition, under the "babies or sunglasses" settings, there will be far more than 300 samples. **I strongly suggest the authors to evaluate their approach following prior arts, instead of just conducting some style transfer experiments on face editing.** **The experiments shown in this submission significantly narrow down the task of "domain adaptation for GANs".**
> > > >
> > > > Hope the above discussions can help explain the change of rating. Also, I suggest ACs to make a thorough discussion on this submission when making the decision. I would also be happy to answer any further questions.

---

> > > > > ### Author Response · Authors · 2021-11-29
> > > > > **FID, Precision, and Recall results**
> > > > >
> > > > > | 1 Shot      | FID | precision     | recall |
> > > > > | :---        |    :----:   |    :----:   |       ---: |
> > > > > | TargetCLIP      |  199.33      |  0  |  0.293   |
> > > > > | Ojha| 158.86        |    0.001   |     0  |
> > > > > | StyleGAN-NADA | 124.55       |    0.118  |     0  |
> > > > > | Ours| 78.35      |   0.326 |     0.017  |
> > > > >
> > > > >
> > > > >
> > > > > | 5 Shot      | FID | precision     | recall |
> > > > > | :---        |    :----:   |    :----:   |       ---: |
> > > > > |FreezeD     |  102.11      | 0.52   |  0.009   |
> > > > > |Ojha |90.42        |   0.19   |     0  |
> > > > > | GenDA | 87.55      |    0.16  |     0.053  |
> > > > >
> > > > >
> > > > >
> > > > > We tried to find other concurrently submitted arXiv papers to get a better understanding of what the reviewer could talk about. We found one single shot method (GenDA) that seems to do something similar to what the reviewer is talking about. We followed the test protocol in this paper to establish improved FID, precision, recall in the following setting: one shot generation of sketches. Our one shot result is even comparable to 5 shot results of other methods (copied from the arXiv paper). We did not run our method as 5 shot method due to time constraints.
> > > > >
> > > > > Even without the new result, we do not think FID, precision, and recall should be the main metric for a comparison. These metrics are not presented in the one shot papers we followed and we would like to argue that the user study we presented is better to establish a contribution.
> > > > >
> > > > > Besides, our visual results were so strong compared to all other methods that this should also be taken into consideration when evaluating the paper.
> > > > >
> > > > > Even though we still would argue that the main point of the new review is a comparison to one or multiple concurrent (but still unnamed) ICLR 2022 submissions, we hope that these new results can convince the reviewer to change the score.

---

> > > > > > ### Comment · Reviewer_d3yx · 2021-11-30
> > > > > > **Thanks for the clarification**
> > > > > >
> > > > > > The additional quantitative results and the comparisons address my major concerns, hence, I would like to raise the rating to 6. Also, I would suggest the authors to report the results in the main paper (or appendix if there are too many contents).

---

### Official Review · Reviewer_Zd5v · 2021-11-02

**Correctness:** 3
**Technical Novelty And Significance:** 3
**Empirical Novelty And Significance:** 3
**Recommendation:** 6
**Confidence:** 4

**Main Review:**

Strengths:
- The proposed method is novel based on best of my knowledge. The paper designed a framework to leverage the prior knowledge in pretrained CLIP, StyleGAN for adaptation. The idea of modelling domain shift as parallel vectors is interesting and seems to be effective.
- The achieves results are very impressive, notably better than competing methods as demonstrated in fig 4. Also due to the latent space is coupled, it can be combined with works in latent code editing (e.g. Styleflow) for attribute manipulation.
- The related work part is well-written and gives a comprehensive review of the related topics.

Weakness:
- Some of the claims are not fully supported by experiments. For e.g. authors claim that "We greatly reduce the mode collapse/overfitting problem which often occurs in one-shot and few-shot domain adaptation.", however I found it's not fully sufficient to support this with only fig 4. Besides the user study, perhaps more quantitative measurements can be reported for ablation study etc.

- The writing can be further refined. Currently the methodology part is a bit difficult to follow and many details seem to be missing. In particular, what do the learnable weights in fig 3 represent? Would the authors elaborate further on which parameters are being learned?

Minor questions:
- For stylegan-nada and few shot gan in fig4, how many reference images are used?


**Summary Of The Paper:**

The paper proposed a method for adapting a trained GAN to another domain using a single image from the target domain. The main technical contributions include a training framework which leverages pretrained CLIP, StyleGAN for adaptation, several new regularizers and other improvements.


**Summary Of The Review:**

Overall I think the paper is interesting, as it proposes a novel way for one-shot GAN adaptation. However I do have some concerns about insufficient experimental validation and clarity in details.

---

> ### Author Response · Authors · 2021-11-22
> **Response to Reviewer Zd5v**
>
> **Q1: Some of the claims are not fully supported by experiments. For e.g. authors claim that "We greatly reduce the mode collapse/overfitting problem which often occurs in one-shot and few-shot domain adaptation.", however I found it's not fully sufficient to support this with only fig 4. Besides the user study, perhaps more quantitative measurements can be reported for ablation study etc.**
>
> R: We will address this by adding an additional figure to the appendix (see Fig. 12). This figure shows the images that result from two prior methods, Gatys et al. (2015)  and AdaIN (Huang & Belongie, 2016) well as a concurrent method called TargetCLIP (Chefer et al 2021). These comparisons were not included in the original submission because we considered StyleGAN-NADA and few-shot GAN adaptation to be state of the art.  For one-shot domain adaptation, it is very difficult to find a quantitative way to measure mode-collapse, for example a human observer can see that the images generated by "few shot" method do not have as much variation as the source images in Domain A, but this is hard to quantify. However, by looking at the results one can verify that our results have much higher visual quality.
>
>
> **Q2: The writing can be further refined. Currently, the methodology part is a bit difficult to follow, and many details seem to be missing. In particular, what do the learnable weights in fig 3 represent? Would the authors elaborate further on which parameters are being learned?**
>
> R: We have added a sentence explaining which weights are learnable in the first paragraph of the 'training' paragraph of section 3. We also explain which layers have learnable parameters in the caption of Fig. 3. Our learnable parameters are the same ones used by StyleGAN-NADA, we train the weights of the StyleGAN generator, except for the mapping and ToRGB layers. Since the $W+$ codes should match in the original and refined GAN, the mapping network must remain the same.
>
>
> **Q3: For stylegan-nada and few shot gan in fig4, how many reference images are used?**
>
> R: In order for the comparison to be fair, we include only methods that can operate on a single reference image. For StyleGAN-NADA, they originally proposed a zero-shot method, however they also demonstrate a one-shot capability - so one image is used in the comparison. Similarly, Few-shot typically uses 10 reference images, however they also claim the ability to do single-shot domain adaptation. We use one reference image in this case as well. In the proposed revision, we did clarify the language in the paper so that readers are not led to believe we use multiple images for the few shot comparison.
>
>
>
>
> **References:**
>
> Chefer, H., Benaim, S., Paiss, R., and Wolf, L. 2021. Image-Based CLIP-Guided Essence Transfer. arXiv preprint arXiv: 2110.12427.
>
> Gatys, L. A., Ecker, A. S., and Bethge, M. 2016. Image style transfer using convolutional neural networks. Proceedings of the IEEE conference on computer vision and pattern recognition, 2414–2423.
>
> Huang, X. and Belongie, S. 2017. Arbitrary Style Transfer in Real-time with Adaptive Instance Normalization. ICCV.

---

### Official Review · Reviewer_eZmY · 2021-11-06

**Correctness:** 4
**Technical Novelty And Significance:** 3
**Empirical Novelty And Significance:** 4
**Recommendation:** 8
**Confidence:** 3

**Main Review:**

**Strengths:**

1. The visualization results demonstrate the proposed method can generate much higher quality images than the baseline StyleGAN-NADA [1], which seems promising to me. The ablation study shows the effect of each regularizer in the loss function.
2. The framework itself is clear and novel.
3. The training and testing are GPU-friendly and the cost time is acceptable.
4. The method is versatile in that it is also equipped with functions like image editing and domain gap controlling.

**Weaknesses:**

1. The biggest problem of the current version of the paper is writing and presentation, especially in Section 3. The writing and presentation is kind of messy to me. Just to mention a few: $v^{\text{ref}}$ first appears on l.9 in Section 3 and later it is defined in equation (1), which means I do not have any idea what $v^{\text{ref}}$ exactly is before reading till equation (1). What is $W+$ space? More details of II2S should be provided for readability.
2. The compared baselines are simply StyleGAN-NADA [1] and few-shot-gan-adaptation [2]. I think there should be more baseline methods in the style transferring area.
3. No quantitative inference time comparison with the baselines are provided.

Some minor points:

(1) Page 5 simply consists of two big figures, leaving much blank space and making the reading experience unsatisfying.
(2) There should be a comma at the end of eq (5).
(3) In the paragraph "Training and Inference Time"  of Section 4, there is something missing in the last sentence "This embedding time is a few seconds using xx".
(4) There is something wrong with the font style of "P-norm" in the paragraph "latent space interpretation and semantic editing".

**References**

[1] Rinon Gal, Or Patashnik, Haggai Maron, Gal Chechik, and Daniel Cohen-Or. Stylegan-nada: Clip-guided domain adaptation of image generators, 2021.
[2] Utkarsh Ojha, Yijun Li, Jingwan Lu, Alexei A. Efros, Yong Jae Lee, Eli Shechtman, and Richard Zhang. Few-shot image generation via cross-domain correspondence, 2021.

**Summary Of The Paper:**

This paper proposes a novel method for single shot domain adaptation for GANs. The approach achieves visually better performance than the current SOTA, allows more degrees of freedom, and is meanwhile efficient to deploy.

**Summary Of The Review:**

My current recommendation is borderline leaning to accept. I may change my score in the rebuttal if my concerns are not addressed well.

--------
The authors addressed most of my concerns and hence I raise my score to 8.

---

> ### Author Response · Authors · 2021-11-22
> **Response to Reviewer eZmY**
>
> **Q1:The biggest problem of the current version of the paper is writing and presentation, especially in Section 3. The writing and presentation is kind of messy to me. Just to mention a few: $v^\text{ref}$  first appears on l.9 in Section 3, and later it is defined in equation (1), which means I do not have any idea what $v^\text{ref}$ exactly is before reading till equation (1). What is $W+$ space? More details of II2S should be provided for readability.**
>
> R: We have moved equation (1) in a revised manuscript so that it is positioned immediately after the introduction of the symbol $v^\text{ref}$.  Our original conception was that the figure and text described the vector, however we understand that the relationship of the vector to the CLIP embeddings can be put closer to its first mention in the text.
>
> With respect to $W+$ space, we will add a sentence to the paper to clarify the relationship between II2S and $W+$ space in the method section.  The $W+$ space is the result of repeating the $W$ space of StyleGAN 18 times, so that different values are used for each block of StyleGAN2. This was shown to lead to improved GAN inversion results by I2S. The II2S method, building on I2S, found that a particular regularization of the GAN-inversion process leads to better results when manipulating latent codes to do image editing. II2S also has a parameter, lambda, so that by increasing lambda we can shift images closer to domain A. This control is not available in e.g. e4e. The similarity of domain-transfer to GAN editing motivated us to use II2S in this work.
>
>
>
> **Q2: The compared baselines are simply StyleGAN-NADA and few-shot-gan-adaptation. I think there should be more baseline methods in the style transferring area.**
>
> R: We have added 3 more baselines in the appendix of the revised paper (see Fig. 12 in the revised rebuttal). Among them, TargetCLIP (Chefer et al. 2021) uses the CLIP loss to find a global editing direction in StyleGAN $W+$ space, Gatys et al (2016)  and AdaIN (Huang & Belongie, 2017)  are single-image based style transfer methods. Note that TargetCLIP is concurrent work, however since it has become available we have also created a comparison against it. Note that our results have much higher visual quality.
>
>
>
> **Q3: No quantitative inference time comparison with the baselines is provided.**
>
> R: Like ours, all the baseline methods in the main paper use StyleGAN as the image generator, so the inference time is the same as ours, which is less than 0.34s at 1024 × 1024 resolution. Similarly, for editing real images, all these baseline methods need to first project the original image into the StyleGAN latent space. Therefore, the time required for embedding is the same, and the embedding process only needs to be performed once.
> While our method uses II2S for training, at inference time any embedding approach (we show e4e in the revised appendix in Figure 13) works nearly as well. During training, only II2S can be used however, because the regularization control is not available.
> We are adding additional baseline methods to a revised manuscript as well, they are TargetCLIP (Chefer et al, 2021) which is concurrent work, Gatys et. al (2015), and AdaIn (Hiang & Belongie,2017).
> Due to the lightweight nature of the network, AdaIN runs at 15 FPS for 512 × 512 images. In contrast, Gatys et al.  requires optimization during inference, so it takes about 43s for each image at 512 × 512 resolution. For more details, please refer to these results:.
>
> Gatys :
> - Time: 46.7s
> - Resolution:  512
>
> AdaIN:
> - Time: 0.06s
> - Resolution:  512
>
> StyleGAN-NADA:
> - Time: 0.34s
> - Resolution: 1024
>
> few-shot-gan-adaptation:
> - Time  0.34s
> - Resolution: 1024
>
> TargetCLIP:
> - Time: 0.34s
> - Resolution: 1024
>
> Ours:
> - Time: 0.34s
> - Resolution: 1024
>
>
>
> **References:**
>
> Chefer, H., Benaim, S., Paiss, R., and Wolf, L. 2021. Image-Based CLIP-Guided Essence Transfer. arXiv preprint arXiv: 2110.12427.
>
> Gatys, L. A., Ecker, A. S., and Bethge, M. 2016. Image style transfer using convolutional neural networks. Proceedings of the IEEE conference on computer vision and pattern recognition, 2414–2423.
>
> Huang, X. and Belongie, S. 2017. Arbitrary Style Transfer in Real-time with Adaptive Instance Normalization. ICCV.

---

> > ### Comment · Reviewer_eZmY · 2021-11-27
> > **Post Rebuttal**
> >
> > Thanks for your detailed reply. I have seen the revision and think the paper has been improved a lot. Hence I raised my score to 8.

---

### Decision · Program_Chairs · 2022-01-20

**Decision:**

Accept (Poster)

**Comment:**

This paper proposes a novel method for the single-shot domain adaptation with the help of Generative Adversarial Nets. The proposed method is interesting, novel, and versatile. Moreover, the performance is impressive and better than the existing methods. However, the writing needs some improvement for better readability. More quantitive results should be provided in the revision for completeness.